# Convergence, plasticity, and tissue residence of regulatory T cell response via TCR repertoire prism

Tatyana O Nakonechnaya[1,2†], Bruno Moltedo[3†], Ekaterina V Putintseva[2†], Sofya Leyn[2], Dmitry A Bolotin[1,2], Olga V Britanova[1,2], Mikhail Shugay[1,2], Dmitriy M Chudakov[1,2,4,5]*

[1]Institute of Translational Medicine, Pirogov Russian National Research Medical University, Moscow, Russian Federation; [2]Genomics of Adaptive Immunity Department, Shemyakin and Ovchinnikov Institute of Bioorganic Chemistry, Moscow, Russian Federation; [3]Howard Hughes Medical Institute and Immunology Program, Sloan Kettering Institute and Ludwig Center at Memorial Sloan Kettering Cancer Center, New York, United States; [4]Central European Institute of Technology, Brno, Czech Republic; [5]Abu Dhabi Stem Cells Center, Abu Dhabi, United Arab Emirates

*For correspondence:
chudakovdm@gmail.com

†These authors contributed equally to this work

Competing interest: The authors declare that no competing interests exist.

**Abstract** Suppressive function of regulatory T cells (Treg) is dependent on signaling of their antigen receptors triggered by cognate self, dietary, or microbial peptides presented on MHC II. However, it remains largely unknown whether distinct or shared repertoires of Treg TCRs are mobilized in response to different challenges in the same tissue or the same challenge in different tissues. Here we use a fixed TCRβ chain FoxP3-GFP mouse model to analyze conventional (eCD4) and regulatory (eTreg) effector TCRα repertoires in response to six distinct antigenic challenges to the lung and skin. This model shows highly 'digital' repertoire behavior with easy-to-track challenge-specific TCRα CDR3 clusters. For both eCD4 and eTreg subsets, we observe challenge-specific clonal expansions yielding homologous TCRα clusters within and across animals and exposure sites, which are also reflected in the draining lymph nodes but not systemically. Some CDR3 clusters are shared across cancer challenges, suggesting a response to common tumor-associated antigens. For most challenges, eCD4 and eTreg clonal response does not overlap. Such overlap is exclusively observed at the sites of certain tumor challenges, and not systematically, suggesting transient and local tumor-induced eCD4=>eTreg plasticity. This transition includes a dominant tumor-responding eCD4 CDR3 motif, as well as characteristic iNKT TCRα CDR3. In addition, we examine the homeostatic tissue residency of clonal eTreg populations by excluding the site of challenge from our analysis. We demonstrate that distinct CDR3 motifs are characteristic of eTreg cells residing in particular lymphatic tissues, regardless of the challenge. This observation reveals the tissue-resident, antigen-specific clonal Treg populations.

## eLife assessment

This manuscript presents a **valuable** approach to exploring CD4+ T-cell response in mice across stimuli and tissues through the analysis of their T-cell receptor repertoires. The authors use a transgenic mouse model with reduced diversity of the T-cell receptor repertoire to elicit more consistent T-cell responses across individuals, demonstrating challenge-specific and tissue-specific responses of regulatory T-cells. The evidence for the authors' conclusions is **solid**, and the work will be of interest to immunologists studying T cell responses.

## Introduction

Treg, which constitute 5–10% of peripheral CD4[+] T cells are indispensable for the maintenance of immunological self-tolerance (*Lee et al., 2012*; *Campbell and Rudensky, 2020*) and regulation of ongoing immune responses to microbiotal (*Xu et al., 2018*; *Suffia et al., 2006*; *Muschaweck et al., 2021*) and foreign (*Bacher et al., 2016*; *Betts et al., 2012*; *Arpaia et al., 2015*; *Popovic et al., 2017*) antigens, contributing to tissue and metabolic homeostasis and regeneration (*Campbell and Rudensky, 2020*). Treg suppresses immune responses through various mechanisms, but primarily does so via antigen-specific (*Levine et al., 2014*) interaction with professional and non-professional antigen-presenting cells (*Mehrfeld et al., 2018*). There is considerable data suggesting that Treg employs higher-affinity T cell receptor (TCRs) and CTLA4 to compete with conventional CD4[+] T cells in an antigen-specific manner (*Tekguc et al., 2021*).

Most Treg cells are produced by the thymus as a functionally distinct FoxP3[+] T cell subset. Thymic Treg is generated based on positive selection against self-peptide/MHCII (pMHCII) complexes in the thymic medulla, where multiple immune cell types may further contribute to this process, including medullary thymic epithelial cells (mTECs), resident and immigrating dendritic cells, macrophages, and B cells (*Campbell and Rudensky, 2020*; *Klein et al., 2014*; *Zegarra-Ruiz et al., 2021*; *Sawant and Vignali, 2014*).

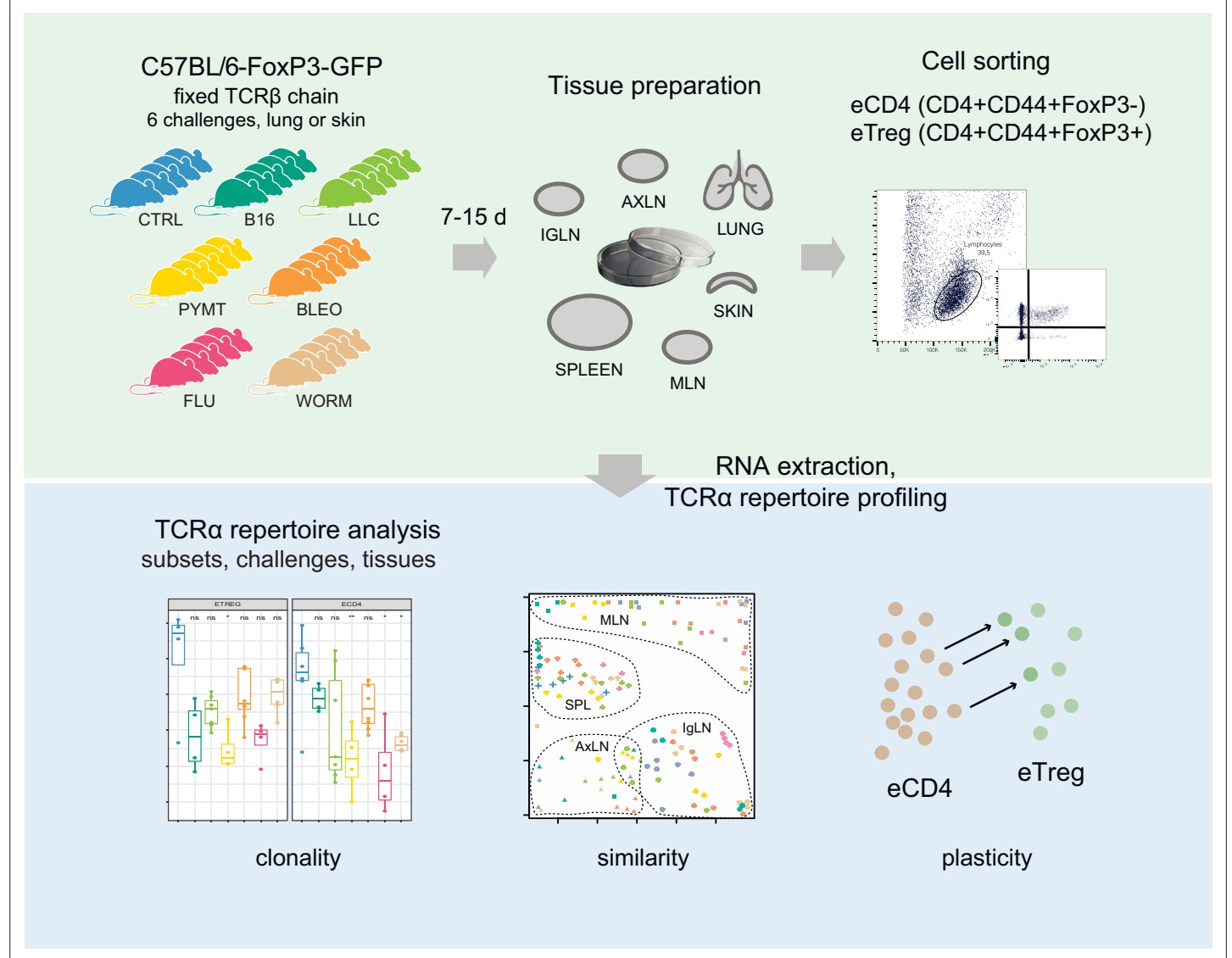

**Figure 1.** Study design.

Here, we investigated the TCR landscape of conventional (eCD4) and regulatory (eTreg) effector CD4⁺ T cells in lungs, spleen, skin, and three lymph node locations after response to six distinct antigenic challenges to the lungs or skin (*Figure 1*). We used FoxP3-GFP mice with a fixed TCRβ context (*Marrack et al., 2017*) and analyzed the TCRα repertoire composition for eCD4 and eTreg cells.

The adaptive immune system in this mouse model relies entirely on TCRα diversity to generate antigen-specific responses. On one hand, the relatively limited TCRα repertoire (*Marrack et al., 2017*) results in highly convergent (*Venturi et al., 2006*) responses that are easily traceable across groups of mice facing the same challenge. The selection of the same or highly similar TCRα sequences against each antigen leads to the formation of clusters of homologous TCRα CDR3s within and between mice. On the other hand, this model allows for the tracking of all challenge-specific or tissue-specific features of the TCR repertoire using only one chain (*Marrack et al., 2017*). Overall, compared to conventional mice, this model provides a more powerful means of monitoring convergent TCR responses. At the same time, the model generally mimics the natural behavior of the full-fledged TCRαβ T cell repertoire.

Both for eCD4 and eTreg subpopulations, our analysis of the TCRα repertoire on the fixed TCRβ background revealed a highly focused response which was distinct for each of the antigenic challenges. The response was clonotypically shared between individual mice and between different tissues and formed clear convergent TCRα CDR3 sequence motifs responding to the challenges.

In conditions with such a highly convergent TCR response involving multiple independently-primed T cell clones with similar TCRα CDR3 sequences, one could expect a similar repertoire of clonotypes to dominate within the eCD4 and eTreg subpopulations, given the assumption that both subsets may start from the same pool of antigen-inexperienced naïve T cells. However, we demonstrate that the TCR landscape of eCD4 and eTreg responses was distinct, suggesting that these effector cell types originate from distinct T cell subsets.

The exceptions were the antigenic challenges with Lewis lung carcinoma (LLC) and MMTV-PyMT-derived tumor cells (PYMT), where we observed increased overlap between the eCD4 and eTreg repertoires at the site of challenge, but not systemically. A dominant eCD4 TCRα motif was also present in corresponding eTreg subsets in the LLC and PYMT challenge.

Similarly, the characteristic innate natural killer T cell (iNKT) TCRα variants (that were abundant within eCD4 subsets upon challenge with LLC, bleomycin, and the helminth *Nippostrongylus brasiliensis*) were exclusively represented in the eTreg subset in the LLC challenge but not in the bleomycin and helminth challenges. We attribute these observations to the transient plasticity of effector CD4⁺ T cells in the context of an immunosuppressive tumor burden.

Furthermore, we reveal tissue-specific eTreg TCRα CDR3 motifs that were always present in specific tissue locations irrespectively of the applied antigenic challenge. This observation highlights the existence of the diverse homing-specific, antigen-specific resident Treg populations.

## Results

### TCRα repertoire sequencing

Six distinct challenges including influenza virus, *N. brasiliensis*, bleomycin-induced injury, LLC, PYMT, and B16 melanoma were applied to the lung and skin of *Foxp3^gfp Tcra^−/+* mice bearing the DO11.10 TCRβ transgene (*Feng et al., 2015*) (3–7 mice per group, see *Supplementary file 1* for details on each tissue and challenge experiment). After incubation, mice were sacrificed and tissue from the lungs, spleen, skin, and three types of lymph nodes (lung-draining mediastinal, MLN, axillary, AXLN, and intraglandular, IGLN) were isolated and digested to generate single-cell suspensions. eCD4 and eTreg CD4⁺ cells were sorted based on FoxP3-GFP signal. RNA-based unique molecular identifier (UMI)-labeled TCRα cDNA libraries were obtained using a previously-reported technique (*Izraelson et al., 2018*; *Logunova et al., 2020*) with minor modifications, and then analyzed using MIGEC (*Shugay et al., 2014*) and MiXCR (*Bolotin et al., 2015*) software. See *Figure 1* for the experimental scheme. Altogether, we sequenced 524 TCRα cDNA libraries, yielding 13,605±14,948 UMI-labeled TCRα cDNA molecules and 3392±3076 TCRα CDR3 clonotypes per eTreg sample, and 37,412±33,129 UMI-labeled TCRα cDNA molecules and 5518±4601 TCRα CDR3 clonotypes per eCD4 sample (see *Supplementary file 2* for details on each cloneset).

Lungs are exposed to a myriad of different insults during their lifetime and are constantly in need of a T cell response to resolve inflammatory insults. Furthermore, Treg with an effector phenotype

accumulates in the inflamed lung (*Fulton et al., 2010*) but it remains poorly understood whether there is a common antigenic denominator driving TCR specificities in the tissue or whether there are selective TCR subsets expanded upon a particular inflammatory insult. In the following analyses, we mainly focused on TCR repertoires obtained from T cells infiltrating the lung tissue, for which we have also obtained the largest collection of samples, with several more specific cross-tissue analyses.

## Clonality of eCD4 and eTreg repertoires in the lung

First, we explored the clonality of eCD4 and eTreg responses in the lungs following distinct antigenic challenges. TCRα repertoire diversity was assessed using several widely used metrics, including observed diversity (number of distinct clonotypes), normalized Shannon Wiener index (repertoire evenness and the extent of clonal expansion), and Chao 1 (estimates lower bound of total diversity based on relative representation of small clonotypes). For normalization, all of these metrics were obtained from datasets that had been down-sampled to 1000 randomly chosen UMI-labeled TCRα cDNAs (*Izraelson et al., 2018*). We observed a prominent clonal response to each of the challenges compared to the control group (*Figure 2a–c*). The decrease in diversity metrics was comparable for the lung-infiltrating eCD4 and eTreg cells. This indicates that the amplitude and focusness of the effector and regulatory T cell response in the lungs are generally comparable.

## Responding eCD4 and eTreg repertoires are distinct

Next, we analyzed the all-versus-all pairwise overlap of the amino acid TCRα CDR3 repertoires of lung-infiltrating eTreg and eCD4 cells. We used F2 similarity metrics in the VDJtools software (*Shugay et al., 2015*), which employs a clonotypewise sum of geometric mean frequencies that takes into account the relative size of shared clonotypes. As such, F2 metrics generally enable comparison of pairs of TCR repertoires with regard to the relative share occupied by common clonotypes. This analysis showed that the eTreg and eCD4 repertoires are highly distinct across distinct challenges (*Figure 2d and e*). eTreg response is convergent and differs for distinct challenges.

We next zoomed in on the eTreg repertoire in order to investigate how focused their antigen-specific response is in the lungs. Repertoire overlap analysis showed that eTreg repertoires were highly similar across mice for each of the challenges, and differed between challenges (*Figure 3a and b*). Cluster sequence analysis further revealed common dominant TCRα CDR3 motifs in response to each of the challenges (*Figure 3a*, *Figure 3—figure supplements 1 and 2*, *Supplementary file 4* and *Supplementary file 5*). It should be noted that eTreg repertoires in the three different cancer challenges (B16, LLC, and PYMT) clustered together and shared some TCRα CDR3 motifs, suggesting a response to shared tumor-associated antigens. In LLC and PYMT, but not in other challenges, some of the characteristic CDR3 motifs were shared with the eCD4 subset.

## eTreg repertoire upon lung challenge is reflected in the draining lymph node

It has previously been shown in a mouse model that the antigen-specific Treg response to influenza in the lungs is also reflected in the lung-draining mediastinal lymph node (MLN) (*Brincks et al., 2013*). Our data revealed this at the repertoire level, showing that the same TCRα CDR3 clonotypes and motifs distinguish antigen-specific eTreg response to each of the different lung challenges in the lung and MLN (*Figure 3b*, *Figure 3—figure supplements 1 and 2*, *Supplementary file 4* and *Supplementary file 5*). It should be noted that each challenge produces its own specific response, wherein eTreg TCRα CDR3 repertoires from the lung and MLN are located side by side. At the same time, the lung and MLN eTreg repertoires of control mice obtained in the absence of any antigenic challenges were distinct, confirming that repertoire similarity is dictated by corresponding antigenic challenges (*Figure 3b*). In contrast, the eTreg repertoires in distant AXLN and IGLN lymph nodes and the spleen were less similar to that observed in the lungs, and clustered separately (*Figure 3c–f*). Altogether, these results demonstrate the selective tissue localization of the antigen-focused Treg response.

## Repertoire focus is the same for the lung and skin tumor localization

The eTreg TCR repertoire in lungs (upon lung tumor challenge) and in the skin (upon corresponding tumor challenge in the skin) was also highly similar (*Figure 4*), demonstrating that the antigen-specific character of the Treg response dominates over the tissue location of the challenge. TCR repertoires

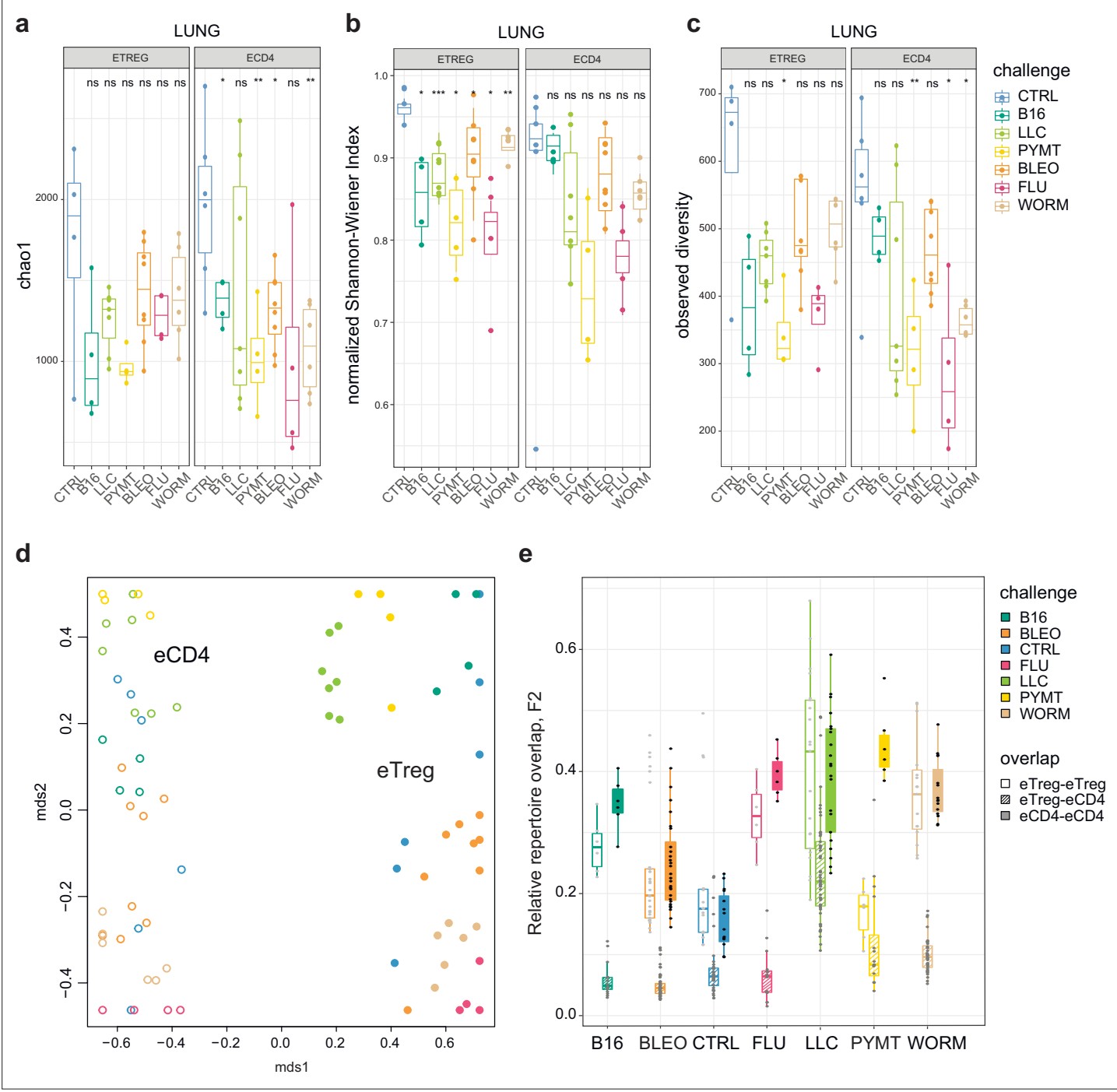

**Figure 2.** Clonality, diversity, and overlap of eTreg and eCD4 subsets. (**a-c**). Clonality and diversity of lung eTreg and eCD4 subsets in response to distinct antigenic challenges. We calculated the (**a**) Chao1 estimator, (**b**) normalized Shannon-Wiener index, and (**c**) observed diversity for each TCRα repertoire obtained from each CD4+ T-cell subset from each animal (3 < n < 7) with different antigenic challenges. p-values are shown as: *p<0.05, **p<0.01, ***p<0.001, and ****p<0.0001, based on parametric t-test for each group versus control. (**d**) Relative overlap between amino acid-defined lung TCRα CDR3 repertoires, visualized as a VDJtools MDS plot. Euclidean distance between points reflects the distance between repertoires. Data were normalized to the top 1000 most frequent clonotypes and weighted by clonotype frequency (F2 metric in VDJtools). Clonotypes were matched on the basis of identical TRAV gene segments and identical TCRα CDR3 sequences. The closer the two circles are, the higher the overall frequency of shared clonotypes. Node colors correspond to the challenge antigen. Filled circles are lung eTreg, open circles are lung eCD4 cells. (**e**) The same F2 metrics as in (**d**), are shown separately for each challenge and for eTreg-eTreg, eTreg-eCD4, and eCD4-eCD4 repertoire overlap. For both (**d**) and (**e**), mice were analyzed in an all-versus-all fashion, irrespective of whether eTreg and eCD4 subsets were obtained from the same or distinct mice. Boxplots show minimum value, the 25th percentile, the median, the 75th percentile and the maximum value.

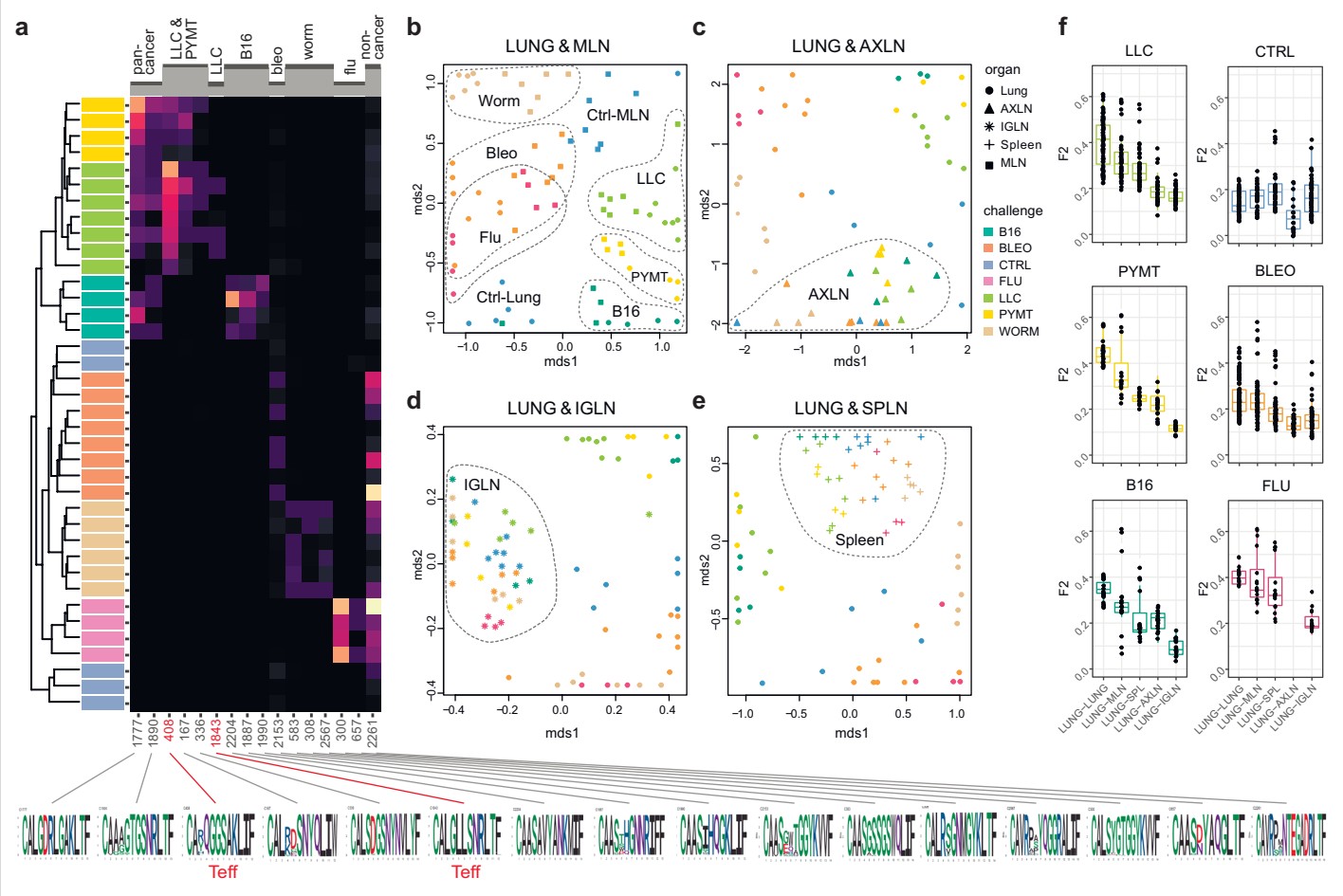

**Figure 3.** eTreg TCR repertoire in various tissues following antigen challenge in the lung. Relative overlap between amino acid-defined eTreg TCRα CDR3 repertoires visualized using VDJtools. Data were normalized to the top 1000 most frequent clonotypes and weighted by clonotype frequency (F2 metric). Clonotypes were matched on the basis of identical TRAV gene segments and identical TCRα CDR3 sequences. (**a**) Dendrogram of Lung eTreg TCRα CDR3 repertoires, where branch length shows the distance between repertoires. Heatmap at right shows the distribution of selected TCRα CDR3 clusters associated with specific challenges. Corresponding TCRα CDR3 logos are shown at the bottom; clusters also observed in corresponding eCD4 samples are indicated in red. (**b–e**) MDS plots comparing repertoires between pairs of tissues, where the Euclidean distance between points reflects the distance between repertoires. Overlap is shown for (**b**) lung versus mediastinal lymph node (MLN), (**c**) lung versus axillary (AXLN), (**d**) lung versus intraglandular (IGLN), and (**e**) lung versus spleen. The closer the two samples are, the higher the overall frequency of shared clonotypes. Node colors correspond to different challenges. Mice were analyzed in an all-versus-all fashion, irrespective of whether tissues were obtained from the same or distinct mice. Circles: lung. Squares: MLN. Triangles: AXLN. Snowflakes: IGLN. Crosses: spleen. (**f**) Graphs show eTreg F2 repertoire overlap between lung tissue from different animals, and between lungs and other tissues of all animals. Boxplots show minimum value, the 25th percentile, the median, the 75th percentile and the maximum value.

The online version of this article includes the following figure supplement(s) for figure 3:

**Figure supplement 1.** TCRα CDR3 clusters behavior in response to different antigenic challenges.

**Figure supplement 2.** TCRα CDR3 cluster motifs.

from eCD4 T cells in the lungs and MLN in response to distinct lung challenges, as well as repertoires of skin eCD4 T cells in response to corresponding skin challenges also clustered together (*Figure 4*), with a magnitude of repertoire convergence that was at the same level as for eTreg cells (*Figure 2e*).

## eCD4 to eTreg conversion is only observed in two cancer challenges

As can be seen in *Figure 2*, TCRα CDR3 eTreg and eCD4 amino acid repertoires in the lungs were more similar in the context of the LLC tumor challenge compared to other challenges, and this could reflect induced plasticity of the eCD4 subset. In order to assess possible eCD4-to-eTreg clonal conversion due to natural helper T cell plasticity, we analyzed eTreg versus eCD4 repertoire overlap at the

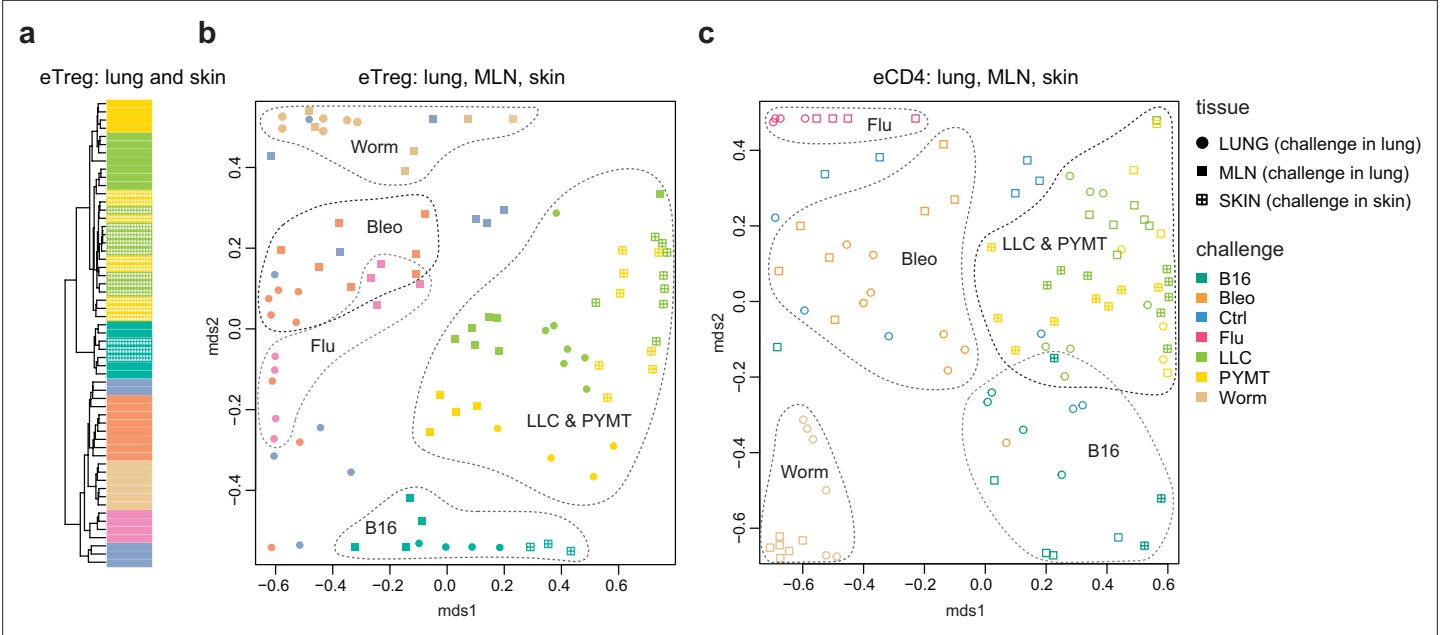

**Figure 4.** Convergence of eTreg and eCD4 TCR response in distinct tissues. (**a,b**) Relative overlap between amino acid-defined lung and mediastinal lymph node (MLN) (upon lung challenge) and skin (upon skin challenge) eTreg TCRα CDR3 repertoires as visualized using VDJtools. Data were normalized to the top 1000 most frequent clonotypes and weighted by clonotype frequency (F2 metric in VDJtools). Clonotypes were matched on the basis of identical TRAV gene segments and identical TCRα CDR3 sequences. (**a**) Dendrogram branch length shows the distance between repertoires. (**b**) Euclidean distance between points on the MDS plot reflects the distance between repertoires. The closer the two samples are, the higher the overall frequency of shared clonotypes. Mice were analyzed in an all-versus-all fashion, irrespective of whether tissues were obtained from the same or distinct mice. Node colors correspond to different challenges. Circles: lung. Crossed squares: skin. (**c**) Relative overlap between amino acid-defined lung and MLN (upon lung challenge) and skin (upon skin challenge) eCD4 TCRα CDR3 repertoires.

level of TCRα CDR3 nucleotide clonotypes for paired eCD4-eTreg samples obtained from the same mice. Estimated eCD4=>eTreg conversion was strongest in the skin and in the lung following LLC and PYMT challenge, but was low or absent in the aftermath of other antigenic challenges, including B16 tumor (***Figure 5a and c***). The effect was probably local and transient, and was not observed at the systemic level in the spleen (***Figure 5b and d***). CDR3a cluster #408 was the major contributor to the observed conversion (***Figure 5e***).

Foxp3[+] Treg-iNKT cells do not naturally arise during development in the thymus, but can be induced and acquire suppressive functions in the periphery under particular pathophysiological conditions (***Moreira-Teixeira et al., 2012***; ***Monteiro et al., 2010***). We specifically observed an increased proportion of Treg-iNKT cells (defined as classic TRAV11-CVVGDRGSALGRLHF-TRAJ18 TCRα) among all eTreg cells upon LLC challenge in lungs (***Figure 5f***). This was not due to an increased presence of eCD4 iNKT cells (***Figure 5g***), suggesting induced iNKT conversion upon LLC lung challenge. The conversion, which was similar to the general conversion shown in ***Figure 5a–d***, was local and was not observed systemically (e.g. in the spleen; ***Figure 5h***).

## Local lymphatic tissue-resident eTreg and eCD4 cells

Global TCRα CDR3 cluster analysis revealed that characteristic eTreg TCR motifs were present in distinct lymphatic tissues, including spleen and thymus, irrespective of the applied challenge (***Figure 3—figure supplements 1 and 2***, ***Supplementary file 4*** and ***Supplementary file 5***). To better illustrate this phenomenon, we performed MDS analysis of TCRα CDR3 repertoires for distinct lymphatic tissues, excluding the lungs due to their otherwise dominant response to the current challenge. This analysis demonstrated the close proximity of eTreg repertoires obtained from the same lymphatic tissues upon all lung challenges and across all animals (***Figure 6a and b***). These results clearly indicate that distinct antigenic specificities are generally characteristic of eTreg cells that preferentially reside in particular lymphatic niches. Notably, the convergence of lymphatic tissue-resident TCR repertoires was less prominent for the eCD4 T cells (***Figure 6c and d***).

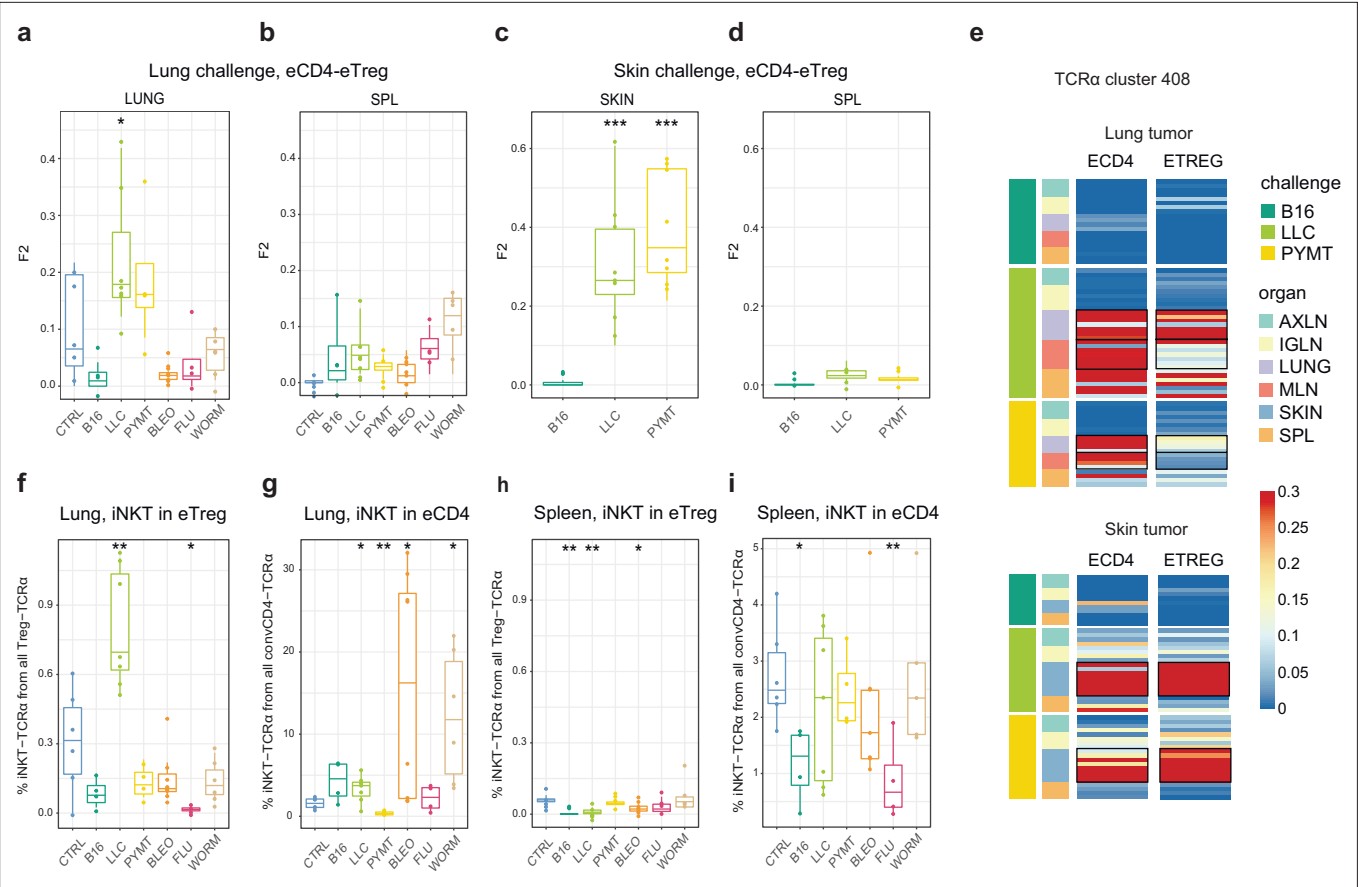

**Figure 5.** Estimated eCD4 to eTreg conversion. (**a–d**) Relative overlap between eTreg and eCD4 nucleotide-defined TCRα CDR3 repertoires upon lung (**a, b**) and skin (**c, d**) challenges. Overlaps are shown for the same organ (**a, c**) and spleen (**c, d**) Data were normalized to the 100 most frequent nucleotide CDR3 clonotypes and overlap was weighted by clonotype frequency (F2 metric). Clonotypes were matched on the basis of identical TRAV and TRAJ gene segments and identical TCRα CDR3 nucleotide sequences. Overlap was analyzed separately for each challenge for eTreg and eCD4 subsets obtained from the same mice, 3 < n < 8. The closer the two samples are, the higher the F2 metric, which reflects the overall frequency of shared clonotypes. (**e**) Distribution of T cell receptor (TCR) cluster 408 in eTreg and eCD4 repertoires upon tumor challenge in lung and skin. (**f**) Treg-iNKT proportion of eTreg cells in lung upon lung challenge. (**g**) Innate natural killer T cell (iNKT) proportion of eCD4 cells in the lung upon lung challenge. (**h**) Treg-iNKT proportion of eTreg cells in spleen upon lung challenge. (**i**) iNKT proportion of eCD4 cells in the spleen upon lung challenge. p-values are shown as: *p<0.05, **p<0.01, ***p<0.001, and ****p<0.0001 based on parametric t-test for each group versus control. Boxplots show minimum value, the 25th percentile, the median, the 75th percentile and the maximum value.

## Discussion

We observed prominent local clonal CD4 + T cell responses to different antigenic challenges within the lung. Remarkably, this effect was comparable for both lung-infiltrating eCD4 and eTreg cells, suggesting similar amplitude and focusness for the effector and regulatory response (**Figure 2a–c**). The TCR repertoires of the eTreg and eCD4 subsets were highly distinct across antigenic challenges (**Figure 2d and e**). eTreg repertoires were highly similar across mice for each challenge (**Figure 3a and b**), and common TCRα CDR3 motifs dominated each response (**Figure 3—figure supplement 1**). eTreg repertoires were similar for the three different cancer challenges, suggesting a response to shared tumor-associated antigens. Antigen-specific eTreg response in lungs was reflected in the lung-draining MLN, but not in distant tissues including the AXLN, IGLN, and spleen (**Figure 3b–e**, **Figure 3—figure supplement 1**), showing the local character of the antigen-specific regulatory response.

For the LLC and PYMT tumors—but not other antigenic challenges—the overall repertoires and dominant CDR3 motifs were shared between the eCD4 and eTreg, suggesting tumor-induced eCD4=>eTreg plasticity. Further analysis showed that eCD4 to eTreg conversion was strongest in the skin and lung following LLC and PYMT tumor challenge, and was observed in the site of challenge

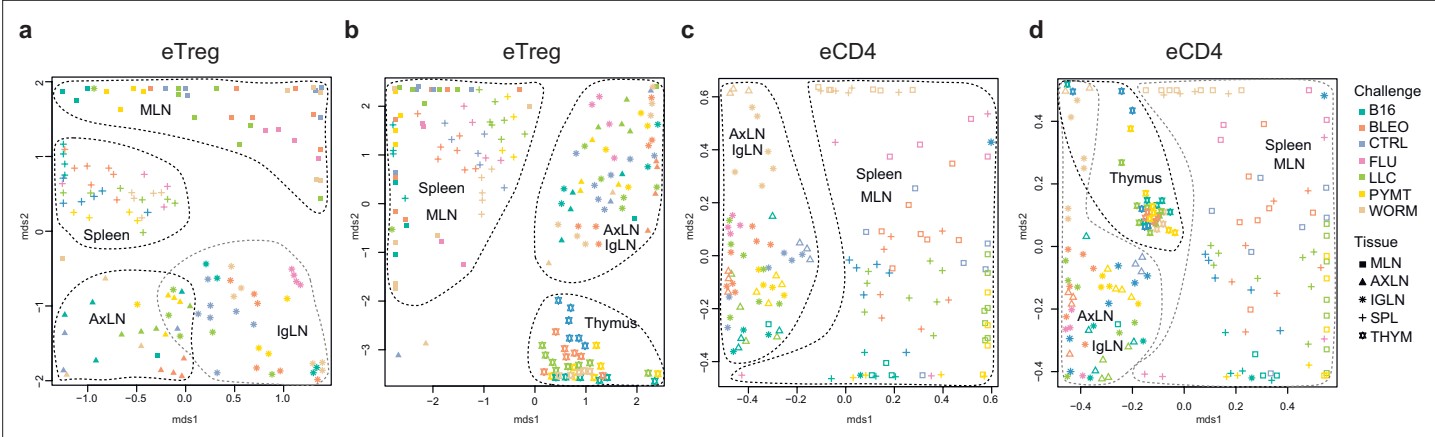

**Figure 6.** Lymphatic tissue-resident eTreg and eCD4 cells. (**a–d**) MDS plots showing the relative overlap between amino acid-defined eTreg TCRα CDR3 repertoires as visualized using VDJtools. Repertoire overlaps are shown for (**a**) eTreg in mediastinal lymph node (MLN), spleen, axillary (AXLN), and intraglandular (IgLN); (**b**) eTreg in MLN, spleen, AXLN, IgLN, and thymus; (**c**) eCD4 in MLN, spleen, AXLN, and IgLN; and (**d**) eCD4 in MLN, spleen, AXLN, IgLN, and thymus. Data were normalized to the top 1000 most frequent clonotypes and weighted by clonotype frequency (F2 metric). Clonotypes were matched on the basis of identical TRAV gene segments and identical TCRα CDR3 sequences. Euclidean distance between points reflects the distance between repertoires; the closer two samples are, the higher is the overall frequency of shared clonotypes. Mice were analyzed in an all-versus-all fashion, irrespective of whether the tissues were obtained from the same or distinct mice. Node colors correspond to the different challenges. Squares: MLN. Triangles: AXLN. Snowflakes: IGLN. Crosses: spleen. Stars: thymus.

but not systemically (*Figure 5a–e*). The presence of an increased proportion of Treg-iNKT cells among the total pool of eTreg cells in the lung upon LLC challenge provided further support for the notion of tumor-induced plasticity (*Figure 5f–h*). This observation is of particular interest in the context of recent reports on innate-like T cell anti-tumor response (*Crowther et al., 2020*).

The local eTreg response to a given tumor type was clonally very similar upon lung and skin challenge, showing that the antigen-specific character of the eTreg response dominates over the tissue location of the initial challenge (*Figure 4*).

Strikingly, when we excluded the site of challenge from the analysis, we observed clear commonality between eTreg repertoires obtained from the same lymphoid tissue in different mice irrespectively of the antigenic challenge (*Figure 6a and b*). This observation directly indicates that distinct antigenic specificities are generally characteristic of Treg cells with distinct tissue residences.

The eCD4 subset generally exhibited similar behavior patterns to the eTregs, but the extent of convergence among eTreg repertoires was even higher in most scenarios compared to the eCD4 subset, indicating the highly antigen-specific nature of the Treg response.

Altogether, our data demonstrate highly antigen-specific and distinct Teff and Treg CD4+ responses to different challenges and support the use of the fixed TCRβ chain FoxP3-GFP mouse as a 'digital' model of TCR response that benefits from access to highly convergent and easy-to-track TCRα CDR3 clusters.

## Methods
### Mice and challenges
C57BL/6 J DO11.10 TCRβ transgenic mice (kindly provided by Philippa Marrack) and crossed to C57BL/6 J *Foxp3*^eGFP *TCRa*^-/- mice. Age-matched DO11.10 TCRβ *Foxp3*^eGFP *TCRa*^+/- +/- were used for experiments. The animals were kept under specific pathogen-free conditions and studied at 6–9 weeks of age.

PyMT cell line was isolated previously from MMTV-PyMT tumor-bearing mice (*Bos et al., 2013*), Lewis Lung Carcinoma (LLC) cell line was a gift of J.Massague (Memorial Sloan-Kettering Cancer Center) and B16-F10 melanoma cells were a gift from C. Ariyan (Memorial Sloan Kettering Cancer Center). All cell lines were grown in DMEM supplemented with 10% FBS, 1% l-glutamine, 1% penicillin-streptomycin, and 10 mM Hepes. Tumor lines were trypsinized and washed with serum-free DMEM. A dose of 1 × 10^5 cells in 200 uL DMEM was injected into each mouse intravenously (i.v) via the tail

vein or subcutaneously (s.c) in the right flank to induce lung or skin tumors, respectively. Influenza A/ Puerto Rico/8/34 (PR8) virus was grown in the cavity of 10 day embryonated chicken eggs, and a dose 100 pfu in 35 uL PBS was used to challenge mice intranasally (i.n) (*Arpaia et al., 2015*). *Nippostrongylus brasiliensis* L3 stage larvae were passaged and isolated from the feces of Wistar rats (*Camberis et al., 2003*). 500–700 infective L3 larvae were injected s.c to mice in 500 uL PBS. Bleomycin for injection was obtained from the pharmacy and each mouse was challenged i.n with a dose of 0.1 U in 35 uL PBS. Control mice were mock-challenged with PBS. Antibodies for magnetic cell isolation and fluorescence-activated cell sorting were purchased from Biolegend, Thermo Fisher. and TONBO BIO. Mouse breeding, challenges, and procedures were performed under protocol 08-10-023 approved by the Sloan Kettering Institute (SKI) Institutional Animal Care and Use Committee.

## Cell isolation, sorting of T cell subsets, and library preparation

To identify and isolate tissue infiltrating T cells and exclude circulating blood lymphocytes, mice from different challenge groups were injected i.v with 0.5 ug of APC anti-CD45 antibody (clone 30-F11) 3 min before euthanasia (*Fan et al., 2018*). Mice bearing lung and skin tumors were euthanized 2 weeks after the challenge and influenza, *Nippostrongylus brasiliensis,* and bleomycin experimental mice at day 9 after challenge.

Lungs from all challenge groups were flushed from excess blood via intracardiac injection with 10 mL of PBS, and each lobe was cut into small pieces and digested with Collagenase A (Roche, 1 mg/mL), DNAse (30 ug/mL) in DMEM 2% FBS for 30 min at 37°C in an orbital shaker. Skin tumors were dissected, minced into small pieces, and digested with Collagenase A as described above for lung samples. Single-cell suspensions were spun and resuspended in sterile EDTA containing FACS buffer (PBS 2% FBS, 1 mM EDTA). Spleens and individual lymph nodes were dissociated with frosted glass slides into single-cell suspensions in FACS buffer. All cell suspensions were filtered with 70 uM strainers (BD Biosciences) and kept on ice until further use.

Individual cell suspensions were enriched for CD4$^+$ T cells using a custom cocktail of biotinylated antibodies (F4/80(clone BM8), anti-mouse I-A/I-E (clone M5/114.15.2), anti-mouse B220(clone RA3-6B2), anti-CD11b (clone M1/70), anti-mouse gamma delta TCR (clone GL3), anti-mouse CD8a (clone 53–6.7), anti-mouse CD11c (clone HL3), anti-mouse Ter119 (clone Ter119), anti-mouse Ly6-G (clone 1A8) at a concentration of 10 ug/mL for each antibody) followed by negative selection with LS columns (Miltenyi). Negative fractions were stained with anti-CD4 (GK1.5, BV605), CD8b (YTS157.7.7, AF700), CD44 (IM7, EF450), CD62L (MEL-14, PE-TexasRed), CD90.2 (clone 30-H12, APC-Cy7), CD45 (clone 30-F11, BV510) and Vb8.2 (clone KJ16-133, PE) antibodies. Effector Treg cells and effector CD4$^+$ T cell subsets were sorted using an Aria-II Cell Sorter (BD Biosciences). Briefly, total effector CD4$^+$ T cells were gated as Vb8.2$^+$ CD4$^+$ CD44$^{hi}$ CD62L$^+$ CD90$^+$ CD45-BV510+CD45 APC- cells. Thereafter, from this gate, conventional effector CD4$^+$ T cells as CD44$^{hi}$ CD62L$^-$ Foxp3-GFP$^-$ cells and effector Treg cells as CD44$^{hi}$ CD62L$^{lo}$ Foxp3-EGFP$^+$ were sorted individually into separate tubes, spun, and lysed in buffer RLT plus (Qiagen) and frozen at –80°C until further use. RNA purification from individual samples (RNeasy Micro Kit, Qiagen), TCR library preparation from cDNA and Library Next Generation Sequencing have been previously described (*Feng et al., 2015*).

## TCR repertoire extraction

Raw 150+150 nt sequence data were analyzed using MIGEC software version 1.2.7. UMI sequences were extracted from demultiplexed data using the Checkout utility. Then, the data were assembled using the erroneous UMI filtering option in the Assemble utility. The minimum required number of reads per UMI was set at two for most tasks. In-frame TCRα and TCRβ repertoires were extracted using MiXCR software (version 3.0.13). Normalization, data transformation, in-depth analyses, and statistical calculations were performed using VDJtools software version 1.2.1 (*Shugay et al., 2015*). R scripts were used to build figures.

## TCRα repertoires diversity analysis

TCRα repertoire diversity was assessed using several widely used metrics, including observed diversity, normalized Shannon-Wiener index, and Chao1. For normalization, all diversity metrics were obtained for datasets that had been downsampled to 1000 randomly chosen, UMI-labeled TCRα cDNA molecules. Samples with UMI <700 were excluded from the analysis.

## TCRα repertoires overlap analysis

Repertoire overlap was analyzed using the weighted F2 (reflecting the proportion of shared T cells between paired repertoires) metric in VDJtools software version 1.2.1. For amino acid overlap metrics calculations, we selected the top 1000 largest clonotypes from each cloneset. Samples with clonotype counts <700 were excluded from the analysis. The top N clonotypes were selected as the top N clonotypes after randomly shuffling the sequences and aligning them in descending order. This was done in order to get rid of the alphabetical order for clonotypes with equal counts (e.g. count = 1 or 2).

## eCD4-to-eTreg conversion analysis

The top 100 clonotypes were extracted from all samples. F2 (weighted on clonotype size) overlaps in terms of nucleotide CDR3 sequence plus identical V and J were calculated for matched pairs of samples as shown in *Supplementary file 3*. Samples with clonotype counts <100 were excluded from the analysis. The top N clonotypes were selected as the top N clonotypes after randomly shuffling the sequences and aligning them in descending order. This was done in order to get rid of the alphabetical order for clonotypes with equal counts (e.g. count = 1 or 2).

## Statistical analysis

Results are shown as mean ± SEM. Statistical analyses were performed on processed datasets in R. Multiple parameter inferences were estimated using a parametric t-test.

## TCRα clusters

TCRα sequence homology clusters were generated using the TCRNET algorithm (*Pogorelyy and Shugay, 2019*). All TCRα CDR3 sequences were pooled, and unique sequences were used to build a graph. Edges of this graph connected CDR3s that differed by a single amino acid mismatch. CDR3s having more neighbors than expected by chance were selected, and connected components from selected CDR3 sequences were used as clusters. As a baseline, we used random mouse TCRα VDJ rearrangements. Each cluster was assigned a frequency based on the total frequency of T cells encoding corresponding CDR3s within a sample. The CDR3 cluster frequency matrix was then subjected to hierarchical clustering using the 'aheatmap' R package with default parameters.

# Acknowledgements

Authors are grateful to Alexander Y Rudensky for his invaluable contribution to the work in all aspects, and to Michael Eisenstein for the helpful edits. Supported by the grant from the Ministry of Science and Higher Education of the Russian Federation 075-15-2019-1789.

# Additional information

### Funding

| Funder | Grant reference number | Author |
| --- | --- | --- |
| Ministry of Science and Higher Education of the Russian Federation | 075-15-2019-1789 | Dmitriy M Chudakov |

The funders had no role in study design, data collection and interpretation, or the decision to submit the work for publication.

### Author contributions

Tatyana O Nakonechnaya, Data curation, Visualization; Bruno Moltedo, Data curation, Investigation, Methodology; Ekaterina V Putintseva, Sofya Leyn, Dmitry A Bolotin, Mikhail Shugay, Data curation; Olga V Britanova, Writing – review and editing; Dmitriy M Chudakov, Conceptualization, Data curation, Supervision, Investigation, Writing – original draft, Project administration, Writing – review and editing

## Author ORCIDs
Olga V Britanova ⬥ http://orcid.org/0000-0002-6295-1392
Mikhail Shugay ⬥ http://orcid.org/0000-0001-7826-7942
Dmitriy M Chudakov ⬥ http://orcid.org/0000-0003-0430-790X

## Ethics
Generation and treatments of mice were performed under protocol 08-10-023 approved by the Sloan Kettering Institute (SKI) Institutional Animal Care and Use Committee. All mouse strains were maintained in the SKI animal facility in specific pathogen free (SPF) conditions in accordance with institutional guidelines and ethical regulations.

Reviewer #1 (Public Review): https://doi.org/10.7554/eLife.89382.3.sa1
Reviewer #3 (Public Review): https://doi.org/10.7554/eLife.89382.3.sa2
Author response https://doi.org/10.7554/eLife.89382.3.sa3

---

## Additional files

### Supplementary files
- Supplementary file 1. Mouse groups metadata.
- Supplementary file 2. Sample metadata.
- Supplementary file 3. eCD4 versus eTreg sample pairing.
- Supplementary file 4. TCR clusters.
- Supplementary file 5. TCR cluster frequency.
- MDAR checklist

### Data availability
All repertoire data used in the manuscript are available on figshare.

The following dataset was generated:

| Author(s) | Year | Dataset title | Dataset URL | Database and Identifier |
|---|---|---|---|---|
| Nakonechnaya T, Chudakov D, Rudensky AY, Putintseva E, Moltedo B | 2024 | Convergence, plasticity, and tissue residence of regulatory and effector T cell response | https://figshare.com/articles/dataset/Convergence_plasticity_and_tissue_residence_of_regulatory_and_effector_T_cell_response/22226155 | figshare, 10.6084/m9.figshare.22226155 |

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
