## [Editor Report · eLife assessment]

This manuscript presents a **valuable** approach to exploring CD4+ T-cell response in mice across stimuli and tissues through the analysis of their T-cell receptor repertoires. The authors use a transgenic mouse model with reduced diversity of the T-cell receptor repertoire to elicit more consistent T-cell responses across individuals, demonstrating challenge-specific and tissue-specific responses of regulatory T-cells. The evidence for the authors' conclusions is **solid**, and the work will be of interest to immunologists studying T cell responses.

---

## [Referee Report · Reviewer #1 (Public Review)]

The authors investigate the alpha chain t cell receptor landscape in conventional vs regulatory CD4 T cells. Overall I think it is a very well thought out and executed study with interesting conclusions. Findings are valuable and are supported by convincing evidence. This work will be of interest for immunologists studying T cells.

Strengths:

- One of a kind evidence and dataset.

- State of the art analyses using well accepted in the literature tools.

- Interesting conclusions on the breadth of immune response to challenges across different types of challenges (tumor, viral and parasitic).

---

## [Referee Report · Reviewer #3 (Public Review)]

This study presents a valuable exploration of CD4+ T cell response in a fixed TCRβ chain FoxP3-GFP mouse model across stimuli and tissues through the analysis of their TCRα repertoires. This is an insightful paper for the community as it suggests several future directions of exploration.

The authors compare Treg and conventional CD4+ repertoires by looking at diversity measures and the relative overlap of shared clonotypes to characterize similarity across different tissues and antigen challenges. They find distinct yet convergent responses with occasional plasticity across subsets for some stimuli. The observed lack of a general behavior highlights the need for careful comparison of immune repertoires across cell subsets and tissues. Such comparisons are crucial in order to better understand the heterogeneity of the adaptive immune response. This mouse model demonstrates its utility for this task due to the reduced diversity of the TCRα repertoire and the ability to track a single chain.

The revised manuscript has significantly improved in terms of clarity of explanations and presentations of the results.

---

## [Author Response]

The following is the authors’ response to the original reviews.

eLife assessmentThis manuscript presents a valuable approach to exploring CD4+ T-cell response in mice across stimuli and tissues through the analysis of their T-cell receptor repertoires. The authors use a transgenic mouse model, in which the possible diversity of the T-cell receptor repertoire is reduced, such that each of a diverse set of immune exposures elicits more detectably consistent T-cell responses across different individuals. However, whereas the proposed experimental system could be utilized to study convergent T-cell responses, the analyses done in this manuscript are incomplete and do not support the claims due to limitations in the statistical analyses and lack of data/code access.

We worked to address the reviewers' concerns below, point-by-point.

All data on immune repertoires are deposited here: https://figshare.com/articles/dataset/Convergence_plasticity_and_tissue_residence_of_regulatory_and_effector_T_cell_response/22226155

We added the Data availability statement to the manuscript.

**Public Reviews:**

**Reviewer #1 (Public Review):**
The authors investigate the alpha chain TCR landscape in conventional vs regulatory CD4 T cells. Overall I think it is a very well thought out and executed study with interesting conclusions. The authors have investigated CDR3 alpha repertoires coupled with a transgenic fixed CDR3beta in a mouse system.Strengths:One of a kind evidence and dataset.State-of-the-art analyses using tools that are well-accepted in the literature.Interesting conclusions on the breadth of immune response to challenges across different types of challenges (tumor, viral and parasitic).

Thank you for the positive view.

Weaknesses:Some conclusions regarding the eCD4->eTreg transition are not so strong using only the data.

The overlaps between the top-nucleotide clones in both LLC and PYMT challenges are prominently above the average, and this result is reproducible in lungs and skin, so we have no doubts based on these data. Further experiments with different methods, including tracking the clonal fates, should clarify and confirm/correct/disprove our findings.

Some formatting issues.

We are working on the manuscript to correct minor errors and formatting.

**Reviewer #2 (Public Review):**
This study investigates T-cell repertoire responses in a mouse model with a transgenic beta chain, such that all T-cells in all mice share a fixed beta chain, and repertoire diversity is determined solely by alpha chain rearrangements. Each mouse is exposed to one of a few distinct immune challenges, sacrificed, and T-cells are sampled from multiple tissues. FACS is used to sort CD4 and Treg cell populations from each sample, and TCR repertoire sequencing from UMI-tagged cDNA is done.Various analyses using repertoire diversity, overlap, and clustering are presented to support several principal findings: (1) TCR repertoires in this fixed beta system have highly distinct clonal compositions for each immune challenge and each cell type, (2) these are highly consistent across mice, so that mice with shared challenges have shared clones, and (3) induction of CD4-to-Treg cell type transitions is challenge-specific.The beta chain used for this mouse model was previously isolated based on specificity for Ovalbumin. Because the beta chain is essential for determining TCR antigen specificity, and is highly diverse in wildtype mice, I found it surprising that these mice are reported to have robust and consistently focused clonal responses to very diverse immune challenges, for which a fixed OVA-specific beta chain is unlikely to be useful. The authors don't comment on this aspect of their findings, but I would think it is not expected *a priori* that this would work.If this does work as reported, it is a valuable model system: due to massively reduced diversity, the TCR repertoire response is much more stereotyped across individual samples, and it is much easier to detect challenge-specific TCRs via the statistics of convergent responses.

This was to some extent expected, since these mice live almost normally and have productive adaptive immune responses and protection. In real life, there are frequent TCR-pMHC interactions where the TCR-alpha chain dominates (https://www.ncbi.nlm.nih.gov/pmc/articles/PMC5701794/; https://pubmed.ncbi.nlm.nih.gov/37047500/). On the fixed TCR-beta background this mechanics starts working full-fledged, essentially substituting TCR-beta diversity, at the extent of relatively simplified TCRab repertoire and probably higher cross-reactivity.

We agree that this is a valuable model, for sure, and indicated this in the last sentence of our Discussion. Now we are also adding this point to the abstract.

While the data and analyses present interesting signals, they are flawed in several ways that undermine the reported findings. I summarize below what I think are the most substantive data and analysis issues.(1) There may be systematic inconsistencies in repertoire sampling depth that are not described in the manuscript. Looking at the supplementary tables (and making some plots), I found that the control samples (mice with mock challenge) have consistently much shallower sampling-in terms of both read count and UMI count-compared with the other challenge samples. There is also a strong pattern of lower counts for Treg vs CD4 cell samples within each challenge.

The immune response of control mice is less extensive, as it should be. Just like the fact that the number of Tregs in tissues is lower than CD4, this is normal. So this all follows the expectations. But please note that we were very accurate everywhere with appropriate data normalisation, using all our previous extensive experience (https://pubmed.ncbi.nlm.nih.gov/29080364/).

In particular (now adding more relevant details to Methods):

For diversity metrics calculations, we randomly sampled an equal number of 1000 UMI from each cloneset. Samples with UMI < 700 were excluded from analysis.

For amino acid overlap metrics calculations, we selected top-1000 largest clonotypes from each cloneset. Samples with clonotype counts < 700 were excluded from analysis.

For nucleotide overlaps metrics calculations (eCD4-eTreg), we selected top-100 clonotypes from each cloneset. Samples with clonotypes < 100 were excluded from analysis.

The top N clonotypes were selected as the top N clonotypes after randomly shuffling the sequences and aligning them in descending order. This was done in order to get rid of the alphabetical order for clonotypes with equal counts (e.g. count = 1 or 2).

Downsampling was carried out using software vdjtools v.1.2.1.

(2) FACS data are not reported. Although the graphical abstract shows a schematic FACS plot, there are no such plots in the manuscript. Related to the issue above, it would be important to know the FACS cell counts for each sample.

Yes, we agree that this is valuable information that should be provided. Unfortunately, this data has not been preserved.

(3) For diversity estimation, UMI-wise downsampling was performed to normalize samples to 1000 random UMIs, but this procedure is not validated (the optimal normalization would require downsampling cells). What is the influence of possible sampling depth discrepancies mentioned above on diversity estimation? All of the Treg control samples have fewer than 1000 total UMIs-doesn't that pose a problem for sampling 1000 random UMIs?

Indeed, I simulated this procedure and found systematic effects on diversity estimates when taking samples of different numbers of cells (each with a simulated UMI count) from the same underlying repertoire, even after normalizing to 1000 random UMIs. I don't think UMI downsampling corrects for cell sampling depth differences in diversity estimation, so it's not clear that the trends in Fig 1A are not artifactual-they would seem to show higher diversity for control samples, but these are the very same samples with an apparent systematic sampling depth bias.

We evaluated this approach through all our work, and summarised in the ref: https://pubmed.ncbi.nlm.nih.gov/29080364/. Altogether, normalising to the same count of randomly sampled UMI seems to be the best approach (although, preferably, the initial sequencing depth should be essentially higher for all samples than the sampling threshold used). Initial sorting of identical numbers of cells and ideally uniform library preparation and sequencing is generally not realistic and does not work in the real world, while UMI downsampling does the same work much better.

(4) The Figures may be inconsistent with the data. I downloaded the Supplementary Table corresponding to Fig 1 and made my own version of panels A-C. This looked quite different from the diversity estimations depicted in the manuscript. The data does not match the scale or trends shown in the manuscript figure.

There was a wrong column for Chao1, now correcting. Also, please note that we only used samples with > 700 UMI. Supplementary Table now corrected accordingly. Also, please note that Figure 1 shows the results for lung samples only.

(5) For the overlap analysis, a different kind of normalization was performed, but also not validated. Instead of sampling 1000 UMIs, the repertoires were reduced to their top 1000 most frequent clones. It is not made clear why a different normalization would be needed here. There are several samples (including all Treg control samples) with only a couple hundred clones. It's also likely that the noted systematic sampling depth differences may drive the separation seen in MDS1 between Treg and CD4 cell types. I also simulated this alternative downsampling procedure and found strong effects on MDS clustering due to sampling effects alone.

That’s right, for the overlap analysis (which values are mathematically proportional to the clonotype counts in both compared repertoires, so the difference in the counts causes major biases) the right way to do it is to choose the same number of clonotypes. See Ref. https://pubmed.ncbi.nlm.nih.gov/29080364/.

We kept only samples with > 700 for the overlap analyses. Some relatively poor samples are present in all challenges, while MDS1 localization has clear reproducible logic, so we are confident in these results.

It is not made clear how the overlap scores were converted to distances for MDS. It's hard to interpret this without seeing the overlap matrix.

This is a built-in feature in VDJtools software (https://pubmed.ncbi.nlm.nih.gov/26606115/).See also here: https://vdjtools-doc.readthedocs.io/en/master/overlap.html.

(6) The cluster analysis is superficial, and appears to have been cherry-picked. The clusters reported in the main text have illegibly small logo plots, and no information about V/J gene enrichments. More importantly, as the caption states they were chosen from the columns of a large (and messier-looking) cluster matrix in the supplementary figure based on association with each specific challenge. There's no detail about how this association was calculated, or how it controlled for multiple tests. I don't think it is legitimate to simply display a set of clusters that visually correlate; in a sufficiently wide random matrix you will find columns that seem to correlate with any given pattern across rows.

Particular CDR3 sequences and VJ segments do not mean much for the results of this manuscript. Logos are given just for visual explanation of how the consensus motifs of the clusters look like.

We now add two more Supplementary Tables and a Supplementary Figure with full information about clusters.

We disagree that the Supplementary Figure 1 (representing all the clusters) looks “messy”. Vice versa, it is surprisingly “digital”, showing the clear patterns of responses and homings. This becomes clear if you visually study it for a while. But yes, it is too big to let the reader focus on this or that aspect. That is why we need to select TCR clusters to illustrate this or that aspect discussed in the work, but they were selected from the overall already structured picture.

(7) The findings on differential plasticity and CD4 to Treg conversion are not supported. If CD4 cells are converting to Tregs, we expect more nucleotide-level overlap of clones. This intuition makes sense. But it seems that this section affirms the consequent: variation in nucleotide-level clone overlap is a readout of variation in CD4 to Treg conversion. It is claimed, based on elevated nucleotide-level overlap, that the LLC and PYMT challenges induce conversion more readily than the other challenges. It is not noted in the textual interpretations, but Fig 4 also shows that the control samples had a substantially elevated nucleotide-level overlap. There is no mention of a null hypothesis for what we'd expect if there was no induced conversion going on at all. This is a reduced-diversity mouse model, so convergent recombination is more likely than usual, and the challenges could be expected to differ in the parts of TCR sequence space they induce focus on. They use the top 100 clones for normalization in this case, but don't say why (this is the 3rd distinct normalization procedure).

Your point is absolutely correct: “This is a reduced-diversity mouse model, so convergent recombination is more likely than usual”. Distinct normalisation procedure was required to focus on the most expanded clonotypes to avoid the tail of (presumably cross-reactive) and identical TCRs present in all repertoires in these limited-repertoire mice. So we downsampled as strictly as possible to minimise this background signal of nucleotide overlap, and only this strict downsampling to the top-100 clonotypes allowed us to visualise the difference between the challenges. This is a sort of too complicated explanation that would overload the manuscript. But your comments and our answers will be available to the reader who wants to go into all the details.

The observed (at this strict downsampling) overlaps between the top-nucleotide clones in both LLC and PYMT challenges are prominently above the average, and this result is reproducible in lungs and skin, so we have no doubts in interpretations based on these data. Further experiments with different methods, including tracking the clonal fates, should clarify and confirm/correct/disprove our findings.

Although interpretations of the reported findings are limited due to the issues above, this is an interesting model system in which to explore convergent responses. Follow-up experimental work could validate some of the reported signals, and the data set may also be useful for other specific questions.

Yes, thank you for your really thorough analysis. We fully agree with your conclusion.

**Reviewer #3 (Public Review):**
Nakonechnaya et al present a valuable and comprehensive exploration of CD4+ T cell response in mice across stimuli and tissues through the analysis of their TCR-alpha repertoires.The authors compare repertoires by looking at the relative overlap of shared clonotypes and observe that they sometimes cluster by tissue and sometimes by stimulus. They also compare different CD4+ subsets (conventional and Tregs) and find distinct yet convergent responses with occasional plasticity across subsets for some stimuli.The observed lack of a general behaviour highlights the need for careful comparison of immune repertoires across cell subsets and tissues in order to better understand their role in the adaptive immune response.In conclusion, this is an important paper to the community as it suggests several future directions of exploration.Unfortunately, the lack of code and data availability does not allow the reproducibility of the results.

Thank you for your positive view.

All data on immune repertoires are deposited here: https://figshare.com/articles/dataset/Convergence_plasticity_and_tissue_residence_of_regulatory_and_effector_T_cell_response/22226155

We added the Data availability statement to the manuscript.

**Recommendations for the authors:**

**Reviewer #1 (Recommendations For The Authors):**
In the manuscript at "yielding 13,369 {plus minus} 1,255 UMI-labeled TCRα cDNA molecules and 3233 {plus minus} 310 TCRα CDR3 clonotypes per sample"I'm not sure how can there be fewer unique DNA molecules than clonotypes in each sample.

That was our mistake for sure, now corrected.

In the manuscript at "This indicates that the amplitude and focused nature of the effector and regulatory T cell response in lungs is generally comparable."I'm not sure it's possible to conclude that a drop in diversity in all conditions necessarily signals a focused nature. Since at this stage, the nature of the colotypes was not compared between conditions, it is not possible to claim a focused nature of the response.

We have softened the wording:

"This could indicate that the amplitude and focused nature of the effector and regulatory T cell response in lungs is generally comparable."

What are your thoughts on why there is such a large overlap between Treg and Teff in the Lung in control? For some replicates it is almost as much as a post-LLC challenge!

There is some natural dispersion in the data, which is generally expectable. The overlaps between the top-nucleotide clones in both LLC and PYMT challenges are prominently above the average, and this result is reproducible in lungs and skin, so we have no doubts based on these data. Further experiments with different methods, including tracking the clonal fates, should clarify and confirm/correct/disprove our findings.

In the manuscript at "These results indicate that distinct antigenic specificities are generally characteristic for eTreg cells that preferentially reside in particular lymphatic niches"I'm not sure we can conclude this from the Figure. Wouldn't you expect the samples to be grouped by color (the different challenges)? Maybe I'm not understanding the sentence!

This is a different story, about resident Tregs, irrespective of the challenge.

The whole explanation is here in the text:

“Global CDR3α cluster analysis revealed that characteristic eTreg TCR motifs were present in distinct lymphatic tissues, including spleen and thymus, irrespective of the applied challenge (Supplementary Fig. 1). To better illustrate this phenomenon, we performed MDS analysis of CDR3α repertoires for distinct lymphatic tissues, excluding the lungs due to their otherwise dominant response to the current challenge. This analysis demonstrated close proximity of eTreg repertoires obtained from the same lymphatic tissues upon all lung challenges and across all animals (Fig. 5a, b). These results indicate that distinct antigenic specificities are generally characteristic for eTreg cells that preferentially reside in particular lymphatic niches. Notably, the convergence of lymphatic tissue-resident TCR repertoires was less prominent for the eCD4 T cells (Fig. 5c, d).”

And in the abstract:

“Additionally, our TCRα repertoire analysis demonstrated that distinct antigenic specificities are characteristic for eTreg cells residing in particular lymphatic tissues, regardless of the challenge, revealing the homing-specific, antigen-specific resident Treg populations. ”

In the manuscript at " Notably, the convergence of lymphatic tissue-resident TCR repertoires was less prominent for the eCD4 T cells ":5b and 5d seem to have the same pattern: Spleen and MLN group together, AxLN and IgLN together and thymus is separate. Do you mean to say that the groups are more diffuse? I feel like the pattern really is the same and it's likely due to some noise in the data…

Yes, we just mean here that eTreg groups are less diffuse - means more convergent.

I'm not sold on the eCD4 to eTreg conversion evidence. Why only limit to the top 100 clones? The top 1000 clones were used in previous analyses! Moreover, the authors claim that calculating relative overlap (via F2) of matching CDR3+V+J genes is evidence of a conversion between eCD4 and eTreg. I think to convince myself of a real conversion, I would track the cells between groups, unfortunately, I'm not sure how to track this.. Maybe looking at the thymus population? For example, what is the overlap in the thymus vs. after the challenge? I don't have an answer on how to verify but I feel that this conclusion is a bit on the weaker end.

Distinct normalisation procedure was required to focus on the most expanded clonotypes to avoid the tail of (presumably cross-reactive) and identical TCRs present in all repertoires in these limited-repertoire mice. So we downsampled as strictly as possible to minimise this background signal of nucleotide overlap, and only this strict downsampling to the top-100 clonotypes allowed us to visualise the difference between the challenges. This is a sort of too complicated explanation that would overload the manuscript. But your comments and our answers will be available to the reader who wants to go into all the details.

The observed (at this strict downsampling) overlaps between the top-nucleotide clones in both LLC and PYMT challenges are prominently above the average, and this result is reproducible in lungs and skin, so we have no doubts in interpretations based on these data. Further experiments with different methods, including tracking the clonal fates, should clarify and confirm/correct/disprove our findings.

There is a nuance in the analysis between Figure 3 and Figure 5 which I think I am not grasping. Both Figures use the same method and the same data but what is different? I think the manuscript would benefit from making this crystal clear. The conclusions will likely be more evident as well!

As explained in the text and above, on Figure 5 “we performed MDS analysis of CDR3α repertoires for distinct lymphatic tissues, excluding the lungs due to their otherwise dominant response to the current challenge.”

The idea of this mini-chapter of the manuscript is to reveal tissue-resident Tregs, distinct for distinct tissues, resident there in all these mice, irrespectively of the challenge we applied. And they are really there (!).

Do the authors plan to share their R scripts?

All calculations were performed in VDJtools. R was only used to build figures. Corrected this in Methods.

Minor typos and formatting issues to address:Typo in Figure 2a the category should read "worm" instead of "warm"

Corrected.

Figure 2a heatmap is missing a color bar indicating the value ranges

The detailed information can be found in additional Supplementary materials.

Figure 2f is never mentioned in the manuscript!

Corrected.

"eTreg repertoire upon lung challenge is reflected in the draining lymph node" - the word upon is of a lower size

Corrected.

The authors should make the spelling of eTreg uniform across the manuscript reg in subscript vs just lower case letters. Same goes for CDR3a vs CDR3\alpha

Corrected.

Figure 4a-d p-values annotations are not shown. Is it because they are not significant?

Corrected.

The spelling of FACS buffer should be uniform (FACs vs FACS, see methods)

Corrected.

In the gating strategy, I would make a uniform annotation for the cluster of differentiation, for example, "CD44 high" vs "CD44^{hi}", pos vs + etc.

Corrected.

Citation for MIGEC software (if available) is missing from methods

Present in the text so probably sufficient.

**Reviewer #2 (Recommendations For The Authors):**
I noticed the data was made available via Figshare in the preprint, but there is no data availability statement in the current ms.

We provided Data availability statement.

The methods state that custom scripts were written to perform the various analyses. Those should be made available in a code repository, and linked in the ms.

All calculations were performed in VDJtools. R was only used to build figures. Corrected this in Methods.

The title mentioned "TCR repertoire prism", so I thought "prism" was the name of a new method or software. But then the word "prism" didn't appear anywhere in the ms.

We just mean viewing or understanding something from a different perspective or through a lens that reveals different aspects or nuances.

Figure 1D lacks an x-axis label.

Worked on the figures in general.

**Reviewer #3 (Recommendations For The Authors):**
The paper is very concise, possibly a bit too much. It could use additional explanations to properly affirm its relevance, for example:why the choice of fixing the CDR3beta background?

To make repertoire more similar across the mice, and to track all the features of repertoire using only one chain.

to what it is fixed?

As explained in Methods:

“C57BL/6J DO11.10 TCRβ transgenic mice (kindly provided by Philippa Marrack) and crossed to C57BL/6J Foxp3eGFP TCRa-/- mice.”

What do you expect to see and not to see in this specific system and why it is important?

As stated above: we expected repertoire to be more similar across the mice, and it is important to find antigen-specific TCR clusters across mice, and to be able to track all the features of the TCR repertoire using only one chain.

Does this system induce more convergent responses? If so, can we extrapolate the results from this system to the full alpha-beta response?

Such a model, compared to conventional mice, is much more powerful in terms of the ability of monitoring convergent TCR responses. At the same time, it behaves natural, mice live almost normally, so we believe it reflects natural behaviour of the full fledged alpha-beta T cell repertoire.

Is the lack of similarity of other tissues to Lung/MLN due to a lack of a response?

As indicated in the title of the corresponding mini-chapter:“eTreg repertoire upon lung challenge is reflected in the draining lymph node”.And conclusion of this mini-chapter is that “these results demonstrate the selective tissue localization of the antigen-focused Treg response. ”

Can you do a dendrogram like 2a for the other tissues to better clarify what is going on there? There is space in the supplementary material.

We built lots of those, but in such single dimension mostly they are less informative compared to 2D MDS plots.

Figure 5 seems a bit out of place as it looks more related to Figure 2. It could maybe be integrated there, sent to supplementary or become Figure 3?

This is a different story, about resident Tregs, irrespective of the challenge.

The whole explanation is here in the text:

“Global CDR3α cluster analysis revealed that characteristic eTreg TCR motifs were present in distinct lymphatic tissues, including spleen and thymus, irrespective of the applied challenge (Supplementary Fig. 1). To better illustrate this phenomenon, we performed MDS analysis of CDR3α repertoires for distinct lymphatic tissues, excluding the lungs due to their otherwise dominant response to the current challenge. This analysis demonstrated close proximity of eTreg repertoires obtained from the same lymphatic tissues upon all lung challenges and across all animals (Fig. 5a, b). These results indicate that distinct antigenic specificities are generally characteristic for eTreg cells that preferentially reside in particular lymphatic niches. Notably, the convergence of lymphatic tissue-resident TCR repertoires was less prominent for the eCD4 T cells (Fig. 5c, d).”

And in the abstract:

“Additionally, our TCRα repertoire analysis demonstrated that distinct antigenic specificities are characteristic for eTreg cells residing in particular lymphatic tissues, regardless of the challenge, revealing the homing-specific, antigen-specific resident Treg populations. ”

Have you explored more systematically the role of individual variability? If you stratify by individual, do you observe any trend? If not this is also an interesting observation to highlight and discuss.

This is inside the calculations and figures/ one dot = 1 mice, so this natural variation is there inside.

Regarding the MDS plots: why are 2 dimensions the right amount? Maybe with 3, you can see both tissue specificity and stimuli contributions. Can you do a stress vs # dimensions plot to check what should be the right amount of dimensions to more accurately reproduce the distance matrix?

Tissue specificity and stimuli contribution is hard to distinguish without focussing on appropriate samples, as we did on Fig. 3 and 5. The work is already not that simple as is, and attempting to analyse this in multidimensional space is far beyond our current abilities. But this is an interesting point for future work, thank you.

Figure 2: A better resolution is needed in order to properly resolve the logo plots at the bottom.

Yes, we worked on Figures, and also provide new Supplementary Figure with all the logos.

No code or data are made available. There is also a lack of supplementary figures that complement and expand the results presented in the main text.

We believe that the main text, although succinct, contains lots of information to analyse and conclusions (preliminary) to make. So we do not see it rational to overload it further.